# Mercury Detection in Benthic and Pelagic Fish Collected from Western Sicily (Southern Italy)

**DOI:** 10.3390/ani9090594

**Published:** 2019-08-22

**Authors:** Gaetano Cammilleri, Francesco Giuseppe Galluzzo, Francesco Fazio, Andrea Pulvirenti, Antonio Vella, Gianluigi Maria Lo Dico, Andrea Macaluso, Gabriele Ciaccio, Vincenzo Ferrantelli

**Affiliations:** 1Food Department, Istituto Zooprofilattico Sperimentale della Sicilia, via Gino Marinuzzi 3, 90129 Palermo, Italy; 2BIOMORF Department, Università degli Studi di Messina, Piazza Pugliatti 1, 98122 Messina, Italy; 3Life Science Department, Università degli studi di Modena e Reggio Emilia, Via Università 4, 41121 Modena, Italy

**Keywords:** contaminants, toxic metals, fish, Mediterranean Sea

## Abstract

**Simple Summary:**

In highly polluted water, fish can accumulate Hg at concentrations that cause risk to human health. This has occurred in Sicily (Southern Italy), where there is activity from the petrochemical pole. In this paper, we present concentrations of mercury in 14 fish species collected from western Sicilian coasts in 2013. A significant difference was found between fish species examined but not between pelagic and benthic fish. Four out of 130 analyzed samples showed mercury concentrations over the European limits.

**Abstract:**

In highly polluted water, fish can accumulate mercury up to a concentration of 10 mgKg^−1^. This has occurred on the eastern coasts of Sicily (Southern Italy), probably due to the intense industrial activity of this area. However, little is known about Hg accumulation in fish of the western Sicilian coasts. In this work, we examined the Hg accumulation of 108 fish samples belonging to 14 species collected from western Sicilian coasts using a direct mercury analyzer. The samples showed a mean mercury concentration of 0.165 ± 0.22 mg kg^−1^ with a maximum in *Lepidopus caudatus* (1.72 mgKg^−1^), exceeding the limits provided by EC Reg. 1881/2006. The lowest Hg levels were found in *Sparus aurata* samples (0.001 mgKg^−1^). A significant difference was found between the fish species examined (*p* < 0.05). The comparison between benthic and pelagic species did not show statistical differences (*p* < 0.05). Fish food constitutes the main route of Hg uptake for humans. Only four of the 130 samples examined reached a mercury concentration over the European limits. The comparative analysis of Hg pollution for benthic and pelagic species did not confirm a different trend in metal contamination.

## 1. Introduction

Mercury (Hg) commonly known as quicksilver, is a heavy, silvery-white liquid metal chemical element, highly toxic to the environment and living creatures [1,2]. Generally, methylation is the most important Hg transformation in an aquatic system [3,4]. Methyl mercury formed by microorganisms is rapidly taken up by living organisms and enters the food chain via plankton filter-feeding bottom invertebrates [4]. Therefore, fish products constitutes the main route of Hg uptake for humans [5,6]. Chronic exposure to Hg may induce sensory disturbances (glove and stocking type), ataxia, dysarthria, constriction of the visual field, auditory disorders, and tremor [7]. In Sicily (Southern Italy), the activity of the Syracusan petrochemical pole resulted in a high total mercury concentration in fish of the east Sicilian coasts [8]. However, little is known about the presence of Hg in fish from the western Sicilian coasts. In this work, the mercury concentration of Mediterranean fish from the western Sicilian coasts was examined by a direct mercury analyzer. The dataset obtained was compared with the European regulation limits (EC Reg. 1881/2006). The correlation between habitats (benthic or pelagic) of each fish species and concentration of mercury was also evaluated. Furthermore, the results obtained in this work were compared with others reported in literature to evaluate possible differences between two various fish areas with different pollution potentialities.

## 2. Materials and Methods

### 2.1. Sampling Plan

A total of 108 fish samples were collected from markets of Palermo from May 2013 to December 2013. All the fish samples came from the north-western coasts of Sicily, as stated on the label (FAO zone 37.1.3). The fish samples included 14 species: *Mullus barbatus, Trachurus trachurus, Sparus aurata, Lepidopus caudatus, Scomber scombrus, Scomber japonicus, Sardina pilchardus, Conger conger, Trachinus draco, Chelidonichthys lucernus, Scorpaena scrofa, Merluccius merluccius, Phycis phycis and Lophius piscatorius.* The samples were filleted with plastic scissors and the muscle tissues were stored at −20 °C until they were analyzed for mercury concentration.

### 2.2. Chemical Analysis

The Hg concentration was determined according to the protocol of Cammilleri et al. (2018) [9]. About 0.1 ± 0.001 g of the samples was weighed, put onto nickel vessels, and introduced to a Milestone DMA-80 Direct Mercury Analyzer (Milestone, Sorisole, Italy). The instrumentation parameters are shown in Table 1. The method was validated for repeatability and expanded measurement uncertainty according to ISO 17025:2005, considering four concentration levels (0.05–2 mgKg^−1^) [2]. The instrumental/method limit of detection and quantification (LOD and LOQ) were assessed by the 3 σ and 10 σ approaches, according to the American Chemical Society committee.

### 2.3. Data Collection and Statistical Analysis

The data obtained were expressed as mg kg^−1^ wet weight (w.w.) and divided by fish species and ecological condition (benthic vs pelagic). The conditions of normal distribution (according to the Shapiro–Wilk test) and homoscedasticity (according to the Levene test) were not met; therefore, a Wilcoxon test was carried out to evaluate significant differences in Hg levels between pelagic and benthic fish samples. Moreover, a Kruskal–Wallis test was carried out to verify significant differences in Hg levels between the fish species examined. The correlation between Hg levels and fish length were examined using Spearman’s rank correlation analysis. All statistical analyses were performed using R version 3.2.2.

## 3. Results

### 3.1. Validation

The validation parameters of the method are listed in Table 2. The method developed was able to determine the Hg concentration of the samples in a wide range of measurements (0.050–2.00 mg kg^−1^) with a mean recovery of 103%.

### 3.2. Hg Content

The Hg content of all the fish samples examined are shown in Table 3. All the samples tested showed Hg content, with mean values of 0.175 ± 0.232 mg kg^−1^, and pelagic and benthic samples showing mean values of 0.1854 ± 0.279 mg kg^−1^ and 0.1576 ± 0.129 mg kg^−1^, respectively. A total of four samples (2.78%) showed Hg values above EC Reg. 1881/2006. Two of these belonged to the *L. caudatus* species, and had Hg content of 1.718 and 1.206 mg kg^−1^. Among pelagic fish, *T. trachurus* samples showed the highest mean Hg content of 0.362 ± 0.182 mg kg^−1^, followed by *E. encrasicolus* and *S. scombrus*. The lowest concentrations were found in *S. aurata* samples (0.033 ± 0.044 mg kg^−1^). Hg content in benthic fish was determined to be in a range between a minimum of 0.005 mg kg^−1^ in *M. barbatus* and a maximum of 1.718 mg kg^−1^ in *L. caudatus*, revealing the highest mean values found in this work (0.388 ± 0.465 mg kg^−1^). Among the benthic fish, the second highest mean value of Hg was found in *C. conger* (0.251 ± 0.139 mg kg^−1^), followed by *T. draco. P. phycis* was the benthic fish that showed the lowest Hg levels (0.066 ± 0.05 mg kg^−1^).

### 3.3. Statistical Analysis

A significant difference in Hg levels was found between fish species (Kruskal–Wallis chi-squared = 46.017, *p*-value = 1.415 × 10^−5^; Figure 1). The post-hoc test showed that *L. caudatus*, *L. piscatorius*, *C. conger*, and *T. trachurus* were the fish species that contributed to the significant difference in Hg levels. However, no significant difference in Hg levels between benthic and pelagic fish samples was found (Wilcoxon *p*-value = 0.1447). No significant correlations were found between fish length and Hg levels of all the fish species examined (*p* > 0.05).

## 4. Discussion

The results of this work showed very low Hg content in fish samples caught off the western Sicilian coasts (FAO zone 37.1.3).This is in accordance with what was reported in literature [10,11,12], which confirms the environmental quality of this fishing area compared to the FAO zone 37.2.2 belonging to the south-eastern Sicilian coasts [8]. Mercury accumulates along the food chain starting from few ppb in algae and plankton and magnifying with relatively higher concentrations in ichthyophagous fish [13,14]. The significant differences in Hg levels between fish species found in this work confirm that the eating habits of fish can have an effect on accumulation [15] regardless of their habitats.

Furthermore, the results demonstrated that the difference in heavy metal accumulation does not apparently depend on fish size, emphasizing the significance of feeding patterns in heavy metal accumulation, according to what was reported in literature [16]. Our findings seem to be in contrast to what was found by Naccari et al. (2015) [10] for fish samples collected from different areas of Sicily, which was a different trend in metal contamination among pelagic, benthic, and demersal species. The highest Hg content was found in *Lepidopus caudatus* samples as a result of their heterogeneous feeding behavior [17]. The diet of *Lepidopus caudatus* is composed of three main items: fish, crustaceans and, to a lesser degree, cephalopods. It is well known that ichthyophagous fish normally have higher levels of mercury than planktivorous fish [15].

However, the Hg levels found in this work are significantly lower than what was found in top pelagic predators of the Mediterranean such as bluefin tuna (*Thunnus thynnus*), rajiformes and sharks—these organisms reached mean Hg concentrations up to two times higher than what was found in this work [9,18]. Farmed *Sparus aurata* samples showed the lowest Hg content, confirming the potential variation in the Hg load of fish muscle between farmed and wild fish in the Mediterranean Sea assumed by Ferreira et al. (2010) [19]. Adult seabass are top predators, capturing shrimps, crabs, squids, and small fishes, and have higher metal concentrations than aquaculture feedingstuffs [20]. Our findings highlight the importance of the food as the main route of metal uptake in sea bream and other farmed fish. However, it should be considered that fishmeal is increasingly substituted in farmed fish diets with ingredients of terrestrial origin, which may affect the mineral content and availability [21]. Among the wild pelagic fish, the sardine samples showed the lowest Hg content, confirming what was reported in other works on small pelagic ichthyofauna [10,11,12].

## 5. Conclusions

The present work confirms the importance of biomonitoring for the toxicological evaluation of fish products [22]. The results obtained from this work confirmed the differences between Sicilian fishing areas regarding the concentration of mercury—the areas belonging to the north-west coasts of Sicily show lower concentrations of mercury, which is a symptom of the excellent ecological status of the area examined. The exposure of the general population to xenobiotic agents through the diet is a major concern for healthcare institutions.

Often the inadequacy of clear information on the extent of exposure and the actual difficulty in extrapolating data on environmental contaminants from humans make it difficult to implement effective preventive regulatory actions. Therefore, the data obtained in this work could be a starting point for further preventive measures aimed at preserving the health of consumers.

## Figures and Tables

**Figure 1 animals-09-00594-f001:**
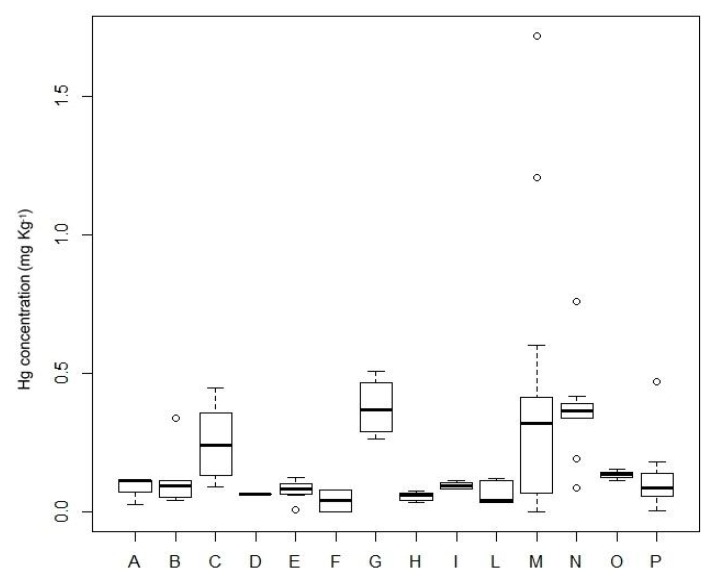
Boxplots of the Hg levels of the fish species examined. Black bars represent the median values; circles represent the outliers. A = *E. encrasicolus*; B = *C. lucernus*; C = *C. conger*; D = *P. phycis*; E = *M. merluccius*; F = *S. aurata*; G = *L. piscatorius*; H = *S. pilchardus*; I = *S. scrofa*; L= *S. scombrus*; M = *L. caudatus*; N = *T. trachurus*; O = *T. draco*; P = *M. barbatus*.

**Table 1 animals-09-00594-t001:** Hg direct analyzer conditions.

Parameter	Value
Combustion temperature (°C)	650
Catalyst temperature (°C)	565
Cuvette temperature (°C)	125
Decomposition temperature (°C)	>600
Start max temperature (°C)	250
Purge time (s)	60
Time for signal registration (s)	30

**Table 2 animals-09-00594-t002:** Results of the validation of the analytical method.

Level	Concentration (mg kg^−1^)	Repeatability (mg kg^−1^)	Expanded Measurement Uncertainty
I	0.1	0.021	±0.020
II	0.5	0.174	±0.153
II	1	0.380	±0.360
IV	2	0.826	±0.710

**Table 3 animals-09-00594-t003:** Hg levels (mean ± SD mg kg^−1^ wet weight (w.w.)) of the fish samples examined. Superscripts ^a,b,c^ mean no significant differences (*p* < 0.05).

Species	*N*	Mean Length (cm)	Hg (Mean ± SD; mg kg ^−1^)	MIN (mg kg ^−1^)	MAX (mg kg ^−1^)
**Pelagic**					
*Engraulis encrasicolus*	3	11.00	0.085 ± 0.049 ^a^	0.027	0.114
*Sardina pilchardus*	12	13.02	0.058 ± 0.015 ^a^	0.0365	0.076
*Scomber scombrus*	6	25.57	0.065 ± 0.041 ^a^	0.035	0.12
*Sparus aurata*	5	24.09	0.033 ± 0.044 ^a^	0.001	0.082
*Trachurus trachurus*	9	20.53	0.362 ± 0.182 ^b^	0.088	0.759
**Total**	**35**		**0.1854 ± 0.279**	**0.001**	**1.718**
**Benthic**					
*Chelidonichthys lucerna*	5	15.35	0.109 ± 0.12 ^a^	0.044	0.337
*Conger conger*	6	54.00	0.251 ± 0.139 ^c^	0.090	0.449
*Mullus barbatus*	18	17.34	0.113 ± 0.101 ^a^	0.005	0.47
*Phycis phycis*	2	39.70	0.066 ± 0.05 ^a^	0.054	0.066
*Scorpaena scrofa*	4	19.86	0.096 ± 0.014 ^a^	0.084	0.113
*Trachinus draco*	3	19.50	0.103 ± 0.02 ^a^	0.115	0.156
**Benthic (benthopelagic)**					
*Lepidopus caudatus*	16	104.07	0.388 ± 0.465 ^b^	0.0003	1.718
*Lophius piscatorius*	4	36.45	0.378 ± 0.109 ^b^	0.263	0.509
*Merluccius merluccius*	15	26.45	0.085 ± 0.030 ^a^	0.009	0.118
**Total**	**73**		**0.1576 ± 0.129**	**0.005**	**0.509**

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
