# Peer review of "Mercury Detection in Benthic and Pelagic Fish Collected from Western Sicily (Southern Italy)"

_animals, 2019, doi:10.3390/ani9090594_

Round 1

Reviewer 1 Report

     Judging from the analysis accuracy shown in Table1, a significant figure thinks figures double at most.  Please inquire. 

     Please specify a wet base or a dry base not only into a table but into text.

     It is known for previous research achievements that mercury concentration has a relation with the age of a fish.  If the scale or otholith which can presume age is left behind, I will recommend you to carry out age determination. Or the relation between size and mercury concentration can also be evaluated. 

     It is important for evaluating the healthy influence by mercury taking in to get to know organic mercury concentration.  If at least the sample whose total mercury concentration was high has organic mercury concentration measured, how is it? 

     This paper treats only the total mercury concentration.  Therefore, the result is not generalizable to heavy metals accumulation. Be careful of this and please revise

Author Response

Dear Reviewer

Please find attached the revised version of our paper. We have carefully revised our manuscript according to your precious suggestions. We have attached a word file with the track changes made to ease your perusal of our manuscript changes.

All the step by step changes are reported below:

Please specify a wet base or a dry base not only into a table but into text.

Dear reviewer, we report this information on the Data collection and statistical analysis section of the main document according to your suggestion.

It is known for previous research achievements that mercury concentration has a relation with the age of a fish. If the scale or otholith which can presume age is left behind, I will recommend you to carry out age determination. Or the relation between size and mercury concentration can also be evaluated.

Dear reviewer, unfortunately, we have not proceeded with the age determination by otholith analysis. Nevertheless, we reported the mean size of each fish species analysed for the evaluation of possible correlations with mercury levels. The results are reported on the main document.

It is important for evaluating the healthy influence by mercury taking in to get to know organic mercury concentration. If at least the sample whose total mercury concentration was high has organic mercury concentration measured, how is it?

Unfortunately, we were not able to analyze the organic mercury levels because of the method used was able only to determine the total mercury concentrations, therefore we cannot satisfy your precious suggestion.

This paper treats only the total mercury concentration. Therefore, the result is not generalizable to heavy metals accumulation. Be careful of this and please revise.

We have made the changes requested on the main document.

Reviewer 2 Report

The tissue or portion (muscle, fillet or whole body) of fish samples used for chemical analysis should be indicated.

Body weight or body length of fish should be indicated.

The concentration of mercury in the muscle of fish is usually correlated with body weight. Figure 1 should be arranged in each fish species. 

Author Response

Dear Reviewer

We have carefully revised our manuscript. We have attached a word file with the track changes made to ease your perusal of our manuscript changes.

All the step by step changes are reported below:

The tissue or portion (muscle, fillet or whole body) of fish samples used for chemical analysis should be indicated.

Dear Reviewer, we reported the samples type in the main document (section 2.1), according to your suggestion.

Body weight or body length of fish should be indicated.

We reported the mean body length of each fish species examined in table 1.

The concentration of mercury in the muscle of fish is usually correlated with body weight.

Dear reviewer, unfortunately, we have not proceeded with the weight determination

Figure 1 should be arranged in each fish species.

We arranged figure 1 according to your suggestion

Reviewer 3 Report

The MS entitled "Mercury detection in benthic and pelagic fish collected from Western Sicily (Southern Italy)" give a comprehensive assessment of Hg burden of fish from Western Mediterranean by the analysis of muscle samples. The novelty of the work relies in the contribution of new epidemiological data on the risk assessment of Mediterranean fish population related to Hg exposure. I also find very interesting the comparison between Benthic and pelagic species. However, I have found some weaknesses in terms of materials and methods and discussion of the results obtained. All my suggestions and comments are listed below:

MATERIALS AND METHODS

AA did not declare if certified reference material were used for the method validation.

AA did not report the statistical methods used for homoschedasticity and normality if distribution of the datasets.

AA should report the formula used for LOD and LOQ detection

Author Response

Dear Rewiever

 We have carefully revised our manuscript according to your precious suggestions. We have attached a word file with the track changes made to ease your perusal of our manuscript changes.

All the step by step changes are reported below:

AA did not declare if certified reference material were used for the method validation.

The certified reference materials was a fish protein certified reference material for trace metals (DORM-4) from the National Research Council of Canada, with a certify quantity of 0.410±0.055.

AA did not report the statistical methods used for homoschedasticity and normality of distribution of the datasets.

The statistical tests used were reported in the Materials and Methods section

AA should report the formula used for LOD and LOQ detection

Dear reviewer, the LOD and LOQ calculation formula used were according to the 3σ and 10σ approaches, according to the ACS committee

Round 2

Reviewer 2 Report

The manuscript was well revised.

Author Response

Dear Reviewer 

Many thanks for your appreciation.

Sincerely